# First Trimester Maternal Homocysteine and Embryonic and Fetal Growth: The Rotterdam Periconception Cohort

**DOI:** 10.3390/nu14061129

**Published:** 2022-03-08

**Authors:** Eleonora Rubini, Katinka M. Snoek, Sam Schoenmakers, Sten P. Willemsen, Kevin D. Sinclair, Melek Rousian, Régine P. M. Steegers-Theunissen

**Affiliations:** 1Department of Obstetrics and Gynecology, Erasmus MC, University Medical Center, 3015 GD Rotterdam, The Netherlands; e.rubini@erasmusmc.nl (E.R.); k.snoek.1@erasmusmc.nl (K.M.S.); s.schoenmakers@erasmusmc.nl (S.S.); m.rousian@erasmusmc.nl (M.R.); 2Department of Biostatistics, Erasmus MC, University Medical Center, 3015 GD Rotterdam, The Netherlands; s.willemsen@erasmusmc.nl; 3School of Biosciences, University of Nottingham, Leicestershire LE12 5RD, UK; kevin.sinclair@nottingham.ac.uk

**Keywords:** embryo, fetus, homocysteine, growth, periconception, prenatal, birth, one carbon metabolism, ultrasound, virtual reality

## Abstract

Homocysteine is a marker for derangements in one-carbon metabolism. Elevated homocysteine may represent a causal link between poor maternal nutrition and impaired embryonic and fetal development. We sought to investigate associations between reference range maternal homocysteine and embryonic and fetal growth. We enrolled 1060 singleton pregnancies (555 natural and 505 in vitro fertilization/intracytoplasmic sperm injection (IVF/ICSI) pregnancies) from November 2010 to December 2020. Embryonic and fetal body and head growth was assessed throughout pregnancy using three-dimensional ultrasound scans and virtual reality techniques. Homocysteine was negatively associated with first trimester embryonic growth in the included population (crown-rump length B −0.023 mm, 95% CI −0.038,−0.007, *p* = 0.004, embryonic volume B −0.011 cm^3^, 95% CI −0.018,−0.004, *p* = 0.003). After stratification for conception mode, this association remained in IVF/ICSI pregnancies with frozen embryo transfer (crown-rump length B −0.051 mm, 95% CI −0.081,−0.023, *p* < 0.001, embryonic volume B −0.024 cm^3^, 95% CI −0.039,−0.009, *p* = 0.001), but not in IVF/ICSI pregnancies with fresh embryo transfer and natural pregnancies. Homocysteine was not associated with longitudinal measurements of head growth in first trimester, nor with second and third trimester fetal growth. Homocysteine in the highest quartile (7.3–14.9 µmol/L) as opposed to the lowest (2.5–5.2 µmol/L) was associated with reduced birth weight in natural pregnancies only (B −51.98 g, 95% CI −88.13,−15.84, *p* = 0.005). In conclusion, high maternal homocysteine within the reference range is negatively associated with first trimester embryonic growth and birth weight, and the effects of homocysteine are dependent on conception mode.

## 1. Introduction

One-carbon metabolism plays an important role in fertility and pregnancy, being involved in developmental programming of cells and tissues, fetal growth and development [1]. Derangements in one-carbon metabolism are represented by elevated homocysteine levels, often in association with folate and vitamin B_12_ deficiency [1]. Such derangements represent one of the causal links between poor maternal nutrition and lifestyle, and the risk of impaired embryonic and fetal development, with implications for long-term postnatal health [2].

One-carbon metabolism is composed of the interlinked folate and methionine cycles, and the transsulfuration pathway, which support essential biological processes including cell multiplication, differentiation, and epigenetic programming [3]. Clinical applicable and usable biomarkers of one-carbon metabolism include homocysteine, folate, and vitamin B_12_. A high blood concentration of homocysteine is recognized as a sensitive marker for general folate deficiency [4,5].

Elevated homocysteine levels are known to impair female fertility and preimplantation embryo development. Studies report that elevated homocysteine impairs gamete quality, decreases fertilization success, and alters epigenetic processes in the preimplantation embryo [2,6]. It is also known that hyperhomocysteinemia in women is associated with pregnancy complications such as recurrent pregnancy loss and pregnancy hypertension disorders, and is negatively associated with birth outcomes such as low birth weight and preterm birth, although there are some inconsistencies in the literature in this regard [2,7,8,9].

Besides birth weight, only a few studies address the association between maternal homocysteine and prenatal development. Parisi et al. reported that high homocysteine levels are associated with delayed embryonic development (Carnegie stages) [10], together with reduced crown-rump length (CRL) and embryonic volume (EV) during the first trimester of pregnancy [11]. In addition, elevated maternal homocysteine represents a risk factor for postnatal neurological impairments in offspring [12]; but few studies investigated fetal brain development and head growth prior to birth [9,13]. Therefore, we hypothesized that maternal homocysteine may negatively influence embryonic and fetal growth throughout pregnancy, including head growth.

We were the first to hypothesize that maternal homocysteine is associated with neural tube defects [14,15] and continued this line of research in humans by revealing associations between maternal one-carbon metabolism and embryonic size and growth [16]. The current study is novel as we investigate associations between maternal homocysteine and serial prenatal size and growth parameters (i.e., embryonic, fetal, and neonatal parameters). Additionally, we limited our analyses of prenatal development to align with the reference range for homocysteine of the hospital laboratory, thereby excluding cases of hyperhomocysteinemia and several adverse pregnancy outcomes, as this has yet to be addressed.

## 2. Materials and Methods

This study is embedded in the Rotterdam Periconception Cohort (Predict Study), a prospective tertiary hospital-based birth cohort study at the Department of Obstetrics and Gynecology of the Erasmus MC, University Medical Center, Rotterdam, The Netherlands. This is an ongoing study initiated in 2009, with the aim of investigating the association between parental periconception health and its impact on reproduction, pregnancy, and neonatal outcomes [17]. The study was approved by the Central Committee on Research in The Hague and the local Medical Ethics Committee of the Erasmus MC (MEC-2004-227).

### 2.1. Study Population

Between November 2010 and December 2020, pregnant women in early first-trimester (<11 weeks of gestation) were eligible for enrolment, provided they (and their partner) had at least reached the age of 18 years, and could read and speak the Dutch language. Figure 1 shows the criteria for inclusion and exclusion for the current study. Women were included if they had conceived naturally or following assisted reproduction (IVF/ICSI). In this study, natural pregnancies also included the use of ovulation induction and intrauterine insemination (IUI) to achieve a pregnancy, as fertilization is achieved via natural occurrence. In all analyses, cases of maternal hyperhomocysteinemia (homocysteine concentration ≥ 15 µmol/L) were excluded to obtain reference range homocysteine concentrations. This occurred in order to exclude any interference on prenatal development from the physiological condition that underlies hyperhomocysteinemia in pregnant women [18,19]. In addition, cases of hyperhomocysteinemia were too low to perform analyses. Exclusion criteria were cases of miscarriage, termination of pregnancy, intrauterine fetal death, stillbirth, postpartum death, twin pregnancy, study withdrawal, fetal congenital abnormalities, and gamete donation for IVF/ICSI pregnancies. All participants gave written informed consent before participation.

### 2.2. Participant Measurements

In naturally conceived pregnancies, gestational age (GA) was calculated from the first day of the last menstrual period of a regular menstrual cycle. For IVF and ICSI pregnancies, GA was calculated from 14 days before the conception date for fresh embryo transfers and 14 days before the fictitious conception date for cryopreserved embryo transfers, adjusted for the age of the embryo at transfer. For pregnancies achieved via IUI, GA was calculated from 14 days before the insemination date. GA was considered unreliable in cases with an unknown last menstrual period or an irregular menstrual cycle (<21 or >35 days). Information on women and their partners was obtained via extensive self-reported questionnaires given at the time of enrolment (<11 weeks of gestation), covering details such as age, anthropometrics, family history, medical history, education, lifestyle behavior, and habits (from 4 weeks before to 8 weeks after conception). Based on Statistics Netherlands, education level was classified as ‘low’ for primary/lower vocational/intermediate secondary school, ‘intermediate’ for higher secondary/intermediate vocational school, and ‘high’ for higher vocational school/university [20]. The consumption of caffeine was defined as ‘none’ for 0 cups of coffee a day, ‘moderate’ for >0.1 and <2 cups of coffee a day, and ‘high’ for ≥2 cups of coffee a day, as 2 cups of coffee contain above the 200 mg/day recommended maximum dose of caffeine in pregnancy.

### 2.3. Laboratory Assays

At the time of enrolment (<11 weeks of gestation), one venous blood sample from both women and their partners was collected for the determination of biomarkers of one-carbon metabolism: serum homocysteine, folate, and vitamin B_12_. Homocysteine was measured until December 2020, whereas folate and vitamin B_12_ were measured until December 2016, due to a change in study protocol. In vitro quantitative determination of total L-homocysteine in human serum was performed using an HCYS assay with a Cobas 8000 system, and in vitro quantitative determination of human serum folate and vitamin B_12_ was performed using the Elecsys Folate III and Elecsys Vitamin B_12_ II assays with a Cobas 8000 system by Cobas^®^ Roche Diagnostics. Protocols are publicly available online on www.diagnostics.roche.com (accessed on 20 December 2021).

### 2.4. Embryonic and Fetal Measurements

To measure embryonic and fetal growth parameters, ultrasound scans were obtained from all women in the first, second, and third trimesters of pregnancy, and neonatal birth weight was obtained from medical records.

#### 2.4.1. First Trimester

Longitudinal transvaginal three-dimensional (3D) ultrasound scans were performed from enrolment to the end of the first trimester of pregnancy with a maximum of 6 scans per pregnancy. 3D ultrasound examinations were performed using a 6–12 MHz transvaginal probe of the Voluson E8 or E10 system (General Electric Medical Systems). 3D ultrasound images were stored as Cartesian volumes and analyzed by virtual reality (VR) systems and V-Scope software (the latter developed at the Erasmus MC in Rotterdam, The Netherlands). The VR systems used in the current study were the BARCO I-Space (until 2019) and Desktop VR system (since 2014), which all use the identical in-house developed software program V-Scope [21,22]. The VR system is a proven and reliable method to measure first trimester embryonic measurements in clinical practice [23]. Measurements of body size (CRL and EV) were performed from 6 + 0 to 13 + 6 weeks of gestation and measured with VR. In a subpopulation of the Predict Study (included population *n* = 118), measurements of head size (head volume (HV), biparietal diameter (BPD), occipitofrontal diameter (OFD), head circumference (HC), and cerebellar size (transcerebellar diameter (TCD)) were performed using an offline 3D software (4D view, GE Medical Systems) [17,24]. Such head and cerebellar measurements were performed at 9 and 11 weeks of gestation because the required reference points to perform such measurements were not visible earlier or only visible in a small proportion of the included population.

#### 2.4.2. Second and Third Trimester

In a subpopulation of the Predict Study (included population *n* = 269) [25], longitudinal transabdominal 2D and 3D ultrasound scans were performed at 22, 26, and 32 weeks of gestation with a maximum of 3 scans per pregnancy. 3D ultrasound examinations were performed using a 4–8 MHz transducer of the Voluson E8 System (General Electric Medical Systems). During ultrasound examination, measurements of fetal biometry (BPD, HC, TCD, abdominal circumference (AC), femur length (FL), and estimated fetal weight (EFW)) were performed. EFW was calculated using the Hadlock formula [26]. Measurement protocols were extensively described in previously published articles from the Department of Obstetrics and Gynecology, Erasmus MC, University Medical Center, The Netherlands [11,21,27,28,29,30].

Additional second trimester measurements of fetal biometry were obtained from medical records and included a single time-point measure of BPD, HC, TCD, AC, FL, and EFW. These measurements were performed during the standard fetal anomaly scan, with a mean of 20 weeks of gestation, according to the ISUOG practice guidelines [31].

### 2.5. Additional Calculations

Birth at <37 weeks of gestation was defined as preterm birth, and neonatal birth weight of ≤2500 g as low birth weight. Hoftiezer percentiles were calculated according to the publication of Hoftiezer et al. [32]. Neonates were considered born small for gestational age (SGA) if Hoftiezer percentile was ≤0.1 and large for gestational age (LGA) if Hoftiezer percentile was ≥0.9. Hoftiezer percentiles represent reference curves from a healthy population.

### 2.6. Statistical Analyses

The Student’s *t*-test with unpaired samples was used to compare continuous variables, Fisher’s exact test for dichotomous categorical variables, and Chi-square test for polychotomous categorical variables. Non-parametric data were log10-transformed to obtain parametric data. Associations were performed with univariate linear regression for continuous exposure and outcome variables, and with binary logistic regression for continuous exposure variables and outcome dichotomous categorical variables.

In subanalyses, homocysteine was divided into categories, as the literature shows that the effect of homocysteine levels on neonatal birth weight is not linear. Homocysteine was divided into fourths with equal number of pregnancies per quartile: Q1 2.5–5.2 µmol/L, Q2 5.3–6.0 µmol/L, Q3 6.1–7.2 µmol/L, Q4 7.3–14.9 µmol/L, and Q1 was used as reference category.

Linear mixed models were used to analyze associations between longitudinal measurements and maternal homocysteine, adjusting for confounders and assuming the effect of homocysteine was constant over gestation. For all analyses, confounders were chosen based on the literature: (Model 1) adjustment for GA and (Model 2) adjustment for GA, conception mode, parity, folic acid supplement use, alcohol use, smoking, maternal BMI, maternal age, and fetal sex. All analyses were stratified by conception mode.

For all analyses, statistical significance was defined as *p* < 0.05. All analyses were executed in IBM SPSS Statistics for Windows (version 25), and R (version 4.1.1).

## 3. Results

### 3.1. Baseline Characteristics of the Study Population

A total of 1060 pregnancies were included in the study, of which 555 (52%) were natural pregnancies and 505 (48%) were conceived by IVF/ICSI, according to the inclusion and exclusion criteria described in Figure 1. The prevalence of hyperhomocysteinemia before the introduction of exclusion criteria was 0.56%. Participants with hyperhomocysteinemia had significantly lower serum folate concentrations when compared to the population with reference range homocysteine (Appendix A). Maternal baseline characteristics of the included and excluded population are shown in Appendix A. The included population was stratified by conception mode and maternal baseline characteristics were compared in Table 1. Women with pregnancies established by IVF/ICSI had lower homocysteine, were older, had a lower BMI, were less likely to drink alcohol and smoke, were 100% compliant with the use of folic acid supplementation and were more likely to be primiparous when compared to women with natural pregnancies. Additionally, serum folate of both the mother and the father were higher in IVF/ICSI pregnancies than in natural pregnancies.

### 3.2. Maternal Homocysteine and Baseline Characteristics

In the included population, women conceiving naturally were more likely to have higher homocysteine levels than women pregnant after IVF/ICSI (OR 0.91 (95% CI 0.84,0.99), *p* = 0.02). Homocysteine was positively associated with maternal BMI (B 1.69 (95% CI 0.46,2.94), *p* = 0.01), with the odds of being a smoker (OR 0.89 (95% CI 0.82,0.97), *p* = 0.01), and with being nulliparous (OR 0.87 (95% CI 0.79,0.97), *p* = 0.01). Maternal homocysteine showed an inverse association with maternal and paternal vitamin B_12_ (B −0.23 (95% CI −0.27,−0.18), *p* < 0.01 and B −0.06 (95% CI −0.11,−0.01), *p* = 0.03 respectively), maternal and paternal serum folate (B −0.08 (95% CI −0.12,−0.04), *p* < 0.01 and B −0.06 (95% CI −0.12,0.00), *p* = 0.04 respectively), and there was a positive association with paternal homocysteine (B 0.09 (95% CI 0.03,0.15), *p* = 0.01).

### 3.3. First Trimester Embryonic Growth

In the included population, the mean GA for CRL and EV was 9 + 6 weeks (69 days). CRL and EV did not differ between natural and IVF/ICSI pregnancies. Linear mixed effect models showed that homocysteine was negatively associated with growth trajectories of CRL and EV in the included overall population (Table 2). The negative association persisted in IVF/ICSI pregnancies with frozen embryo transfer but not in IVF/ICSI pregnancies with fresh embryo transfer and natural pregnancies after stratification for conception mode (Table 2). In the included population, the same association was found if CRL and EV were analyzed at 7, 9, and 11 weeks of gestation individually (Appendix A). At single time-point measurements, the negative association with CRL and EV was present for all modes of conception (Appendix A).

In the subpopulation including 118 pregnancies, embryonic BPD, OFD, HV, HC, and TCD at 9 and 11 weeks of gestation did not differ between natural and IVF/ICSI pregnancies. Linear mixed effect models showed that maternal homocysteine was not associated with longitudinal embryonic head growth (Appendix A).

### 3.4. Second and Third Trimester Fetal Growth

Fetal BPD, TCD, AC, FL and EFW at a mean of 20 weeks of gestation did not differ between natural and IVF/ICSI pregnancies, but HC was significantly smaller in IVF/ICSI pregnancies when compared to natural pregnancies (mean 175.7 mm vs. 176.91 mm, *p* = 0.03). Linear regression showed that homocysteine was not associated with second trimester fetal growth (Table 3). Similarly, no association was found between second trimester fetal growth and quartiles of homocysteine.

In the subpopulation including 269 pregnancies, longitudinal fetal BPD, HC, TCD, AC, FL, and EFW measurements were obtained at a mean of 27 weeks of gestation (range 20–34 weeks of gestation). These measurements did not differ significantly between conception modes. Associations between homocysteine and longitudinal fetal BPD, HC, TCD, AC, and EFW were not significant, except for FL in IVF/ICSI pregnancies (Appendix A).

### 3.5. Birth Weight

In the included population, the mean GA at birth was 39 + 1 weeks, and the average birth weight was 3286 g. Birth weight did not differ between natural and IVF/ICSI pregnancies. Linear regression showed that homocysteine was not associated with birth weight but was negatively associated with Hoftiezer percentiles in natural pregnancies (Appendix A). The direction of association was opposite for mode of conception, as a positive association was found for IVF/ICSI pregnancies with fresh or frozen embryo transfer. We also found that homocysteine in the highest quartile, compared to the lowest quartile, was associated with reduced birth weight (Table 4) and Hoftiezer percentiles in natural pregnancies only (Appendix A). Homocysteine did not influence the odds of having SGA and LGA neonates (Appendix A), nor the odds of being born preterm or having a low birth weight.

## 4. Discussion

The present study sought to investigate the association between maternal first trimester homocysteine levels, within the standard reference range, and longitudinal embryonic and fetal growth from the sixth week of gestation until birth, based on mode of conception. We demonstrated that maternal homocysteine is negatively associated with embryonic growth (CRL, EV) during the first trimester, specifically in IVF/ICSI pregnancies with frozen embryo transfer. Embryonic head growth was not affected by maternal homocysteine levels. Homocysteine was not associated with second and third trimester fetal growth. Levels of homocysteine in the highest quartile were associated with reduced birth weight and Hoftiezer percentiles in natural pregnancies only.

Homocysteine is an important biological and clinical marker of one-carbon metabolism. It is well recognized that high homocysteine levels are indicative of folate and/or vitamin B_12_ deficiency [2,3], and this strong inverse association was consistent with our findings. In addition, the positive association we found between homocysteine, BMI, and smoking has been reported previously [33,34,35]. Women with a high BMI often follow a nutritiously poor diet and suffer from micronutrient deficiencies, explaining a higher risk of high homocysteine levels [34]. Smoking is associated with alterations of one-carbon metabolites within the liver, as well as the altered expression of enzymes involved in the one-carbon metabolism [36]. We report higher homocysteine levels and similar aforementioned associations in natural pregnancies, as opposed to IVF/ICSI pregnancies. Indeed, women undergoing ART treatment are most dedicated to pursue a healthy lifestyle, and their compliance to folic acid supplement may explain their lower homocysteine levels [37]. In line with this, we report that changes in homocysteine levels within our population are mostly a reflection of lifestyle and health-related habits, which is also confirmed by the positive association between maternal and paternal homocysteine.

Deficiencies of one-carbon substrates and co-factors during the periconceptional period are involved in alterations of the embryo’s genetic and epigenetic landscape, which negatively impact prenatal and early postnatal offspring growth [6]. Elevated maternal homocysteine is negatively associated with embryonic and fetal growth. Previous findings from our research group show that homocysteine is negatively associated with first trimester embryonic development according to the Carnegie stages in IVF/ICSI pregnancies only [10]. Here, we confirm results from our previous findings from Parisi et al., which was performed in a smaller study population [11]. However, we found that homocysteine was negatively associated with embryonic growth trajectories in IVF/ICSI pregnancies with frozen embryo transfer only. This is in agreement with studies which found that early embryonic growth patterns differ between frozen and fresh cycles [38], and children born from frozen embryos are at increased risk of being large for gestational age [39,40]. Potential epigenetic modifications during freezing and thawing may make embryos from frozen cycles more vulnerable to alterations of the maternal environment. In conclusion, first trimester embryonic growth in IVF/ICSI pregnancies may be most vulnerable to the effects of maternal homocysteine.

One-carbon metabolism is also involved in brain development and neural programming, and derangements of the pathway increase the short-term risk of embryonic neural tube-related defects and long-term childhood behavior, cognition, and autism spectrum disorders [41]. As reported in our recent review, elevated homocysteine is associated with adult cognitive impairment, but much less is known about its role in prenatal head growth and development [41]. Here, we show that homocysteine is not associated with head growth at any stage of pregnancy. These results are consistent with studies that found that homocysteine is not associated with neonatal HC [9,13,42,43]. However, we have included a small study group with restricted measurements of head and brain structures, so we cannot discount the possibility that homocysteine may have an effect on prenatal brain growth. Lastly, the postnatal effects that homocysteine may exert on brain activity may not be reflected by prenatal growth, or may develop postnatally only; this functional aspect requires further investigation.

First trimester embryonic growth is associated with second and third trimester fetal growth, and consequently with neonatal weight at birth [27]; therefore, it was unexpected to find no influence of homocysteine on fetal growth from the second trimester onwards. Previous studies reported that only homocysteine ≥ 8.31 µmol/L (compared to 5.81–6.60 µmol/L) was associated with reduced fetal FL and EFW during late pregnancy. However, it remains unclear whether these results would be consistent if homocysteine was analyzed continuously [44]. Our observations consider homocysteine as a continuous variable, but we also compared the effects of high versus low levels of homocysteine. With this approach, we show that the merits of homocysteine as a clinical biomarker on mid- and late pregnancy fetal growth may be limited. Fetal growth is dependent on placenta function, which plays a major role from mid-pregnancy. Placental function is disrupted by hyperhomocysteinemia [45], which generates reactive oxygen species such as hydrogen peroxide which lead to oxidative stress, endothelial dysfunction, and placenta-related clotting disorders; all of which can limit placental and fetal blood supply and, consequently, fetal growth [46,47]. As cases with hyperhomocysteinemia were excluded, the likelihood of finding any negative effects of homocysteine on placenta function and consequently fetal growth was small.

This study adds to the current debate on homocysteine and birth weight. Elevated homocysteine is associated with SGA neonates, preterm birth, and low neonatal birth weight [9,44,48,49,50]. However, the findings in these studies applied specifically to homocysteine levels in the highest category (usually >8 µmol/L, including cases of hyperhomocysteinemia) when compared to the lowest category (usually <3 µmol/L). These findings are in agreement with our results, in which homocysteine in the highest quartile (7.3–14.9 µmol/L) reduced birth weight and Hoftiezer percentiles in natural pregnancies only, although no cases of hyperhomocysteinemia were included. Instead, when homocysteine was analyzed as a continuous variable, no effect on birth weight was found. In secondary analyses, we included the three cases of hyperhomocysteinemia eligible after exclusion criteria (homocysteine 16.30–24.90 µmol/L) and found the same associations, with larger effect sizes for birth weight and Hoftiezer percentiles (data not shown). Homocysteine may serve as a clinical biomarker of impaired birth weight in natural pregnancies only. High levels of homocysteine within the reference range may still represent a risk factor for low birth weight.

Despite selecting women with a ‘healthy’ reference range of homocysteine, we found that there was still a negative association with embryonic growth during the first trimester. This observation may be due to epigenetic regulation of embryonic development; cell lineage specification and tissue patterning are predominant during the first trimester, under the influence of one-carbon metabolism [51]. Similarly, high levels of homocysteine within our reference range had a negative association with birth weight. This evidence suggests that high levels of homocysteine, even within the reference range, may represent a pregnancy risk factor.

Additionally, we confirm that conception mode is a strong determinant of the effects homocysteine exerts on first trimester embryonic growth and neonatal birth weight. Embryos from IVF/ICSI pregnancies with frozen embryo transfer appear to be most sensitive to the negative effects of homocysteine during the first trimester, whereas embryos from natural pregnancies were most vulnerable to negative effects at birth. This may suggest that ART treatment factors specific to the in vitro culture environment, and/or freezing/thawing procedures, may influence the way the embryo responds to the surrounding maternal environment during development. ART treatment may alter the embryo’s epigenome, which therefore responds differently to the influence of homocysteine during pregnancy.

This study is the first to analyze, longitudinally, embryonic, and fetal growth from the first trimester until birth. Detection of early pregnancy intrauterine growth is currently advantageous, as the majority of studies include birth as the earliest measurable time point. With this approach, we were able to keep track of offspring development throughout pregnancy, knowing that growth patterns are GA-dependent. All included longitudinal measurements have previously shown to have excellent inter- and intra-observer reproducibility [17]. The same fetal measurements at late gestation were analyzed both at single and multiple time points, using different measurement approaches, which reinforced the reliability of the results. Lastly, the VR approach to perform 3D embryonic measurements substantially improved the preciseness of measurements as opposed to 2D measurements.

The study design is a tertiary cohort, which includes a high-risk population that may introduce bias in the results and reduce external validity. A population-based cohort may avoid such interferences [52]. Moreover, we adjusted for confounders based on published knowledge and, as a result, residual confounding could be considered as analysis limitation. The sample size of the subpopulation was also a limitation. Growth measurements are a reflection of gross anatomy and have limited usefulness when trying to understand underlying molecular and biological processes. Animal studies would provide complementary insights in this regard [2,3,6].

Reduced embryonic and fetal growth is a sign of disturbed development and can lead to adverse postnatal growth patterns and development with long-lasting effects during the life course [53,54]. For this reason, reduced embryonic growth patterns in the first trimester and low birth weight, associated with elevated homocysteine exposure, may represent a risk factor for adverse postnatal effects. Here we confirm that homocysteine is a relevant biomarker for periconceptional health conditions in clinical practice. Caregivers should be aware that homocysteine is a sensitive marker for maternal health status, particularly with regards to nutrition (B vitamin deficiencies) and lifestyle [2]. We therefore recommend routine analysis of homocysteine levels in preconceptional and pregnant women and their partners, to treat such imbalances in a timely manner, as it also has been reported that high homocysteine levels in men affect sperm quality and DNA methylation [55,56]. If homocysteine is found to be elevated during pregnancy, extra fetal ultrasound examinations should be considered to aid decision making regarding obstetric surveillance (e.g., if growth restriction or changes in growth patterns are detected). Based on our findings, we recommend that special attention should be given to women who require ART treatment and who undergo a frozen cycle. Treatment could be postponed in cases of elevated maternal (and paternal) homocysteine levels. In combination with this, preconceptional and pregnant women should be encouraged to follow a healthy diet and lifestyle, to reduce the risks of elevated homocysteine levels during the periconception period and beyond. This can be achieved via tailored and ‘blended’ lifestyle care (e.g., smarterpregnancy.co.uk, accessed on 20-12-2021) [57,58,59,60,61,62].

Future research should investigate the role of homocysteine in women undergoing ART treatment to identify which factors related to fertility or the ART treatment itself influence the embryo’s response to maternal one-carbon metabolism. Lastly, during pregnancy, the reference range of homocysteine should be redefined or adapted, as high levels of homocysteine within the current reference range appear to negatively impact prenatal growth.

## Figures and Tables

**Figure 1 nutrients-14-01129-f001:**
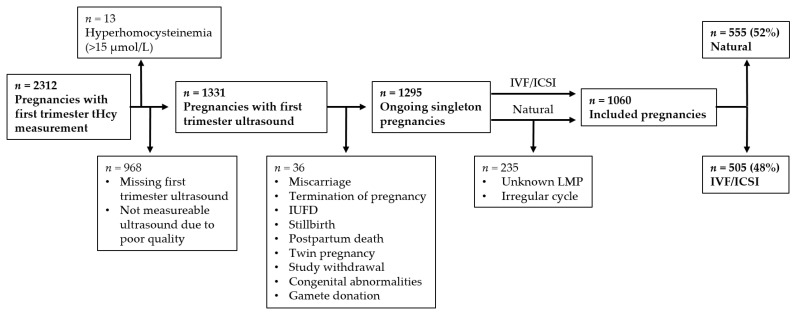
Flowchart of included and excluded population (2010–2020) based on predefined criteria. Abbreviations: ICSI, intracytoplasmic sperm injection; IUFD, intrauterine fetal death; IVF, in vitro fertilization; LMP, last menstrual period; tHcy, total homocysteine.

**Table 1 nutrients-14-01129-t001:** Maternal baseline characteristics of natural and IVF/ICSI pregnancies of the included population.

**Characteristics**	**Natural Pregnancies** **(*n* = 555)**	**IVF/ICSI Pregnancies** **(*n* = 505)**	***p*-Value**
Age ^a^, mean ± SD	31.8 ± 4.4	33.1 ± 4.3	**0.000**
Ethnicity, *n* (%)			0.553
Dutch	424 (80)	390 (81)	
Western, other	27 (5)	31 (6)	
Non-western	77 (15)	63 (13)	
Education level, *n* (%)			**0.042**
Low	46 (9)	25 (5)	
Intermediate	172 (33)	181 (38)	
High	309 (59)	274 (57)	
BMI ^b^, median (IQR)	24.5 (22.2–28.5)	24.4 (21.9–27.7)	**0.007**
Nulliparous, *n* (%)	131 (25)	249 (51)	**0.000**
Alcohol use, *n* (%)	197 (37)	110 (23)	**0.000**
Smoking, *n* (%)	86 (16)	57 (12)	**0.047**
Folic acid supplement use, *n* (%)	520 (98)	484 (100)	**0.002**
Caffeine use, *n* (%)			0.285
None	231 (44)	225 (46)	
Moderate	223 (42)	181 (37)	
High	76 (14)	79 (16)	
tHcy ^e^, mean ± SD	6.5 ± 1.7	6.3 ± 1.4	**0.020**
**Biomarkers ***	**Natural Pregnancies** **(*n* = 411)**	**IVF/ICSI Pregnancies** **(*n* = 271)**	***p*-Value**
tHcy ^e^, median (IQR)	6.2 (5.3–7.2)	6.0 (5.2–6.9)	0.055
Serum folate ^c^, median (IQR)	38.9 (31.9–44.7)	41.0 (34.9–55.5)	**0.002**
Vitamin B_12_ ^d^, median (IQR)	316.5 (237.8–397.0)	316.0 (256.0–404.0)	0.238
Paternal tHcy ^e^, median (IQR)	11.7 (10.0–13.8)	11.3 (9.7–13.3)	0.509
Paternal serum folate ^c^, median (IQR)	17.5 (13.9–22.1)	18.9 (15.0–24.2)	**0.005**
Paternal vitamin B_12_ ^d^, median (IQR)	344.0 (271.0–419.0)	323.5 (260.0–416.0)	0.426

^a^ years, ^b^ kg/m^2^, ^c^ nmol/L, ^d^ pmol/L, ^e^ µmol/L. * Subgroup from November 2010 to December 2016. Alcohol use, smoking, folic acid supplementation and caffeine use were measured during the periconception period. Parametric data is represented as mean ± SD, whereas non-parametric data is represented as median (IQR). Abbreviations: BMI, body mass index; IUI, intrauterine insemination; IVF, in vitro fertilization; ICSI, intracytoplasmic sperm injection; SD, standard deviation; IQR, interquartile range. *p*-value < 0.05 represented in bold.

**Table 2 nutrients-14-01129-t002:** Association between maternal homocysteine and longitudinal first trimester embryonic CRL and EV in the included population (*n* = 1060), natural pregnancies (*n* = 555) and IVF/ICSI pregnancies (*n* = 505).

	Model 1		Model 2	
	CRL			
	Beta (95% CI) √mm	*p*-value	Beta (95% CI) √mm	*p*-value
Included population *n* = 919	**−0.027 (−0.043, −0.012)**	**<0.001**	**−0.023 (−0.038, −0.007)**	**0.004**
Natural pregnancies *n* = 494	**−0.027 (−0.052, −0.001)**	**0.039**	−0.019 (−0.044, 0.006)	0.141
IVF/ICSI pregnancies, FrET *n* = 143	**−0.051 (−0.080, −0.022)**	**<0.001**	**−0.051 (−0.081, −0.023)**	**<0.001**
IVF/ICSI pregnancies, FET *n* = 282	−0.015 (−0.031, 0.001)	0.061	−0.013 (−0.029, 0.002)	0.095
	EV			
	Beta (95% CI) ^3^√cm	*p*-value	Beta (95% CI) ^3^√cm	*p*-value
Included population *n* = 898	**−0.013 (−0.021, −0.006)**	**<0.001**	**−0.011 (−0.018, −0.004)**	**0.003**
Natural pregnancies *n* = 482	**−0.014 (−0.026, −0.002)**	**0.022**	−0.011 (−0.022, 0.001)	0.079
IVF/ICSI pregnancies, FrET *n* = 144	**−0.024 (−0.039, −0.009)**	**0.001**	**−0.024 (−0.039, −0.009)**	**0.001**
IVF/ICSI pregnancies, FET *n* = 272	−0.005 (−0.013, 0.003)	0.213	−0.004 (−0.012, 0.003)	0.305

CRL = crown-rump length, EV = embryonic volume, FET = fresh embryo transfer, FrET = frozen embryo transfer. Model 1: adjusted for GA, Model 2: adjusted for GA, conception mode (included population only), parity, smoking, folic acid supplement use, alcohol, BMI, age, and fetal sex. *p*-value < 0.05 represented in bold.

**Table 3 nutrients-14-01129-t003:** Association between maternal homocysteine and fetal growth from the second trimester fetal anomaly scan in the included population (*n* = 1060), natural pregnancies (*n* = 555), and IVF/ICSI pregnancies (*n* = 505).

	Model 1		Model 2	
	BPD			
	Beta (95% CI) mm	*p*-value	Beta (95% CI) mm	*p*-value
Included population *n* = 966	−0.002 (−0.099, 0.095)	0.966	0.009 (−0.092, 0.111)	0.855
Natural pregnancies *n* = 519	0.025 (0.704, −0.104)	0.704	0.042 (−0.100, 0.183)	0.563
IVF/ICSI pregnancies *n* = 447	−0.007 (−0.154, 0.140)	0.925	−0.004 (−0.149, 0.141)	0.955
	HC			
	Beta (95% CI) mm	*p*-value	Beta (95% CI) mm	*p*-value
Included population *n* = 982	−0.067 (−0.327, 0.193)	0.613	−0.035 (−0.304, 0.234)	0.801
Natural pregnancies *n* = 525	−0.027 (−0.385, 0.330)	0.881	0.003 (−0.381, 0.386)	0.989
IVF/ICSI pregnancies *n* = 457	0.009 (−0.351, 0.369)	0.961	0.054 (−0.297, 0.406)	0.762
	TCD			
	Beta (95% CI) mm	*p*-value	Beta (95% CI) mm	*p*-value
Included population *n* = 972	−0.025 (−0.058, 0.008)	0.137	−0.023 (−0.057, 0.012)	0.198
Natural pregnancies *n* = 519	−0.044 (−0.087, 0.000)	0.051	−0.040 (−0.088, 0.008)	0.100
IVF/ICSI pregnancies *n* = 453	0.027 (−0.020, 0.074)	0.256	0.020 (−0.027, 0.067	0.409
	AC			
	Beta (95% CI) mm	*p*-value	Beta (95% CI) mm	*p*-value
Included population *n* = 980	0.011 (−0.276, 0.297)	0.942	0.053 (−0.250, 0.357)	0.731
Natural pregnancies *n* = 524	0.136 (−0.240, 0.511)	0.478	0.149 (−0.262, 0.561	0.477
IVF/ICSI pregnancies *n* = 456	−0.002 (−0.462, 0.423)	0.994	0.048 (−0.386, 0.481)	0.829
	FL			
	Beta (95% CI) mm	*p*-value	Beta (95% CI) mm	*p*-value
Included population *n* = 980	−0.041 (−0.110, 0.027)	0.235	−0.036 (−0.112, 0.019)	0.345
Natural pregnancies *n* = 525	−0.030 (−0.121, 0.061)	0.521	−0.018 (−0.121, 0.085)	0.729
IVF/ICSI pregnancies *n* = 455	−0.018 (−0.119, 0.082)	0.722	−0.031 (−0.139, 0.077)	0.569
	EFW			
	Beta (95% CI) g	*p*-value	Beta (95% CI) g	*p*-value
Included population *n* = 835	−0.845 (−2.356, 0.665)	0.272	−0.614 (−2.249, 1.021)	0.461
Natural pregnancies *n* = 450	−0.160 (−2.204, 1.883)	0.878	0.099 (−2.215, 2.414)	0.933
IVF/ICSI pregnancies *n* = 385	−0.666 (−2.715, 1.384)	0.524	−0.665 (−2.767, 1.455)	0.541

AC = abdominal circumference, BPD = biparietal diameter, EFW = estimated fetal weight, FL = femur length, HC = head circumference, TCD = transcerebellar diameter. Fetal measurements were performed at a mean 20 weeks of gestation. Model 1: adjusted for GA, Model 2: adjusted for GA, conception mode (included population only), parity, smoking, folic acid supplement use, alcohol, BMI, age, and fetal sex.

**Table 4 nutrients-14-01129-t004:** Association between quartiles of homocysteine and birth weight in the included population (*n* = 983), natural pregnancies (*n* = 529), and IVF/ICSI pregnancies (*n* = 454).

		Model 1		Model 2	
	Included population				
		Beta (95% CI) g	*p*-value	Beta (95% CI) g	*p*-value
*n* = 229	Q1	Reference		Reference	
*n* = 233	Q2	**−83.97 (−161.0.7, −6.88)**	**0.030**	−69.017 (−148.84, 10.81)	0.090
*n* = 297	Q3	−14.03 (−51.73, 23.68)	0.465	−12.27 (−50.09, 25.56)	0.524
*n* = 224	Q4	−13.02 (−40.44, 14.40)	0.351	−15.97 (−44.26, 12.29)	0.267
	Natural pregnancies				
		Beta (95% CI) g	*p*-value	Beta (95% CI) g	*p*-value
*n* = 127	Q1	Reference		Reference	
*n* = 111	Q2	−62.38 (−171.91, 47.15)	0.263	−96.74 (−208.45, 14.96)	0.090
*n* = 155	Q3	−37.75 (−88.45, 12.95)	0.144	−50.29 (−103.09, 2.49)	0.060
*n* = 136	Q4	−33.62 (−68.39, 1.14)	0.060	**−51.98 (−88.13, −15.84)**	**0.005**
	IVF/ICSI pregnancies				
		Beta (95% CI) g	*p*-value	Beta (95% CI) g	*p*-value
*n* = 102	Q1	Reference		Reference	
*n* = 122	Q2	−92.28 (−202.91, 18.36)	0.102	−42.04 (−160.02, 75.93)	0.483
*n* = 142	Q3	14.33 (−42.72, 71.37)	0.621	23.02 (−32.36, 78.39)	0.414
*n* = 88	Q4	14.51 (−30.35, 59.36)	0.524	28.44 (−17.81, 74.70)	0.226

Model 1: adjusted for GA, Model 2: adjusted for GA, conception mode (included population only), parity, smoking, folic acid supplement use, alcohol, BMI, age, fetal sex. Quartiles of homocysteine are represented as Q1 2.5–5.2 µmol/L, Q2 5.3–6.0 µmol/L, Q3 6.1–7.2 µmol/L, Q4 7.3–14.9 µmol/L, and Q1 was used as reference category. *p*-value < 0.05 represented in bold.

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
