# Peer review of "First Trimester Maternal Homocysteine and Embryonic and Fetal Growth: The Rotterdam Periconception Cohort"

_nutrients, 2022, doi:10.3390/nu14061129_

Round 1

Reviewer 1 Report

In this manuscript, the authors investigated the association between maternal first trimester homocysteine levels within reference range and longitudinal embryonic and fetal growth based on mode of conception. The research design was appropriate and the results were meaningful. However, there were still some questions to be solved before acceptance.

  1. Many studies have demonstrated that elevated homocysteine impacted embryonic development. What's the novelty of this article? It is suggested to be clearly stated in the introduction.
  2. One-carbon metabolism included many essential metabolites, such as methionine, serine, glycine, folate. Why did you choose to detect vitamin B12, the cofactor in one-carbon metabolism?
  3. Did homocysteine affect fetal development by modulating one-carbon metabolism? Or by other means?

Author Response

Many studies have demonstrated that elevated homocysteine impacted embryonic development. What's the novelty of this article? It is suggested to be clearly stated in the introduction.

Thank you for the suggestion. We have re-phrased the end of the introduction to emphasize the novelty and added value of this study compared to the current literature. The text we modified is the following (lines 67-74): ‘We were the first to hypothesize that maternal homocysteine is associated with neural tube defects [14, 15] and continued this research line in humans by showing associations between the maternal one-carbon metabolism and embryonic size and growth [16]. This study is novel as we investigate associations between maternal homocysteine and serial prenatal size and growth parameters (i.e. embryonic, fetal and neonatal parameters). Additionally, we limit our analyses of prenatal development to align with the reference range for homocysteine of the hospital laboratory, thereby excluding cases of hyperhomocysteinemia and several adverse pregnancy outcomes, as this has yet to be addressed.

One-carbon metabolism included many essential metabolites, such as methionine, serine, glycine, folate. Why did you choose to detect vitamin B12, the cofactor in one-carbon metabolism?

Thank you for the question. Studies show that among the metabolites of the one-carbon metabolism, the most clinically relevant for pregnancy are folate and vitamin B12. Our study shows an inverse relationship between folate/vitamin B12 and homocysteine, which was sufficient to demonstrate derangements of the one-carbon metabolism pathway. Folate levels give us an indication of the amount of substrates available for the methionine cycle, and vitamin B12 levels inform us on the likelihood of such substrates being converted. Indeed, vitamin B12 deficiency leads to lower levels of methionine synthase, which causes functional folate deficiency by ‘trapping’ an increased proportion of 5-methyl folate derivatives (the so-called ‘folate trap’). Folate and vitamin B12 seem sufficient to determine how much homocysteine is actually stored or re-converted to methionine. Furthermore, folate and vitamin B12 deficiencies are associated with adverse prenatal development and pregnancy-related disorders, whereas not much has been documented on the other metabolites. According to the literature, folate and vitamin B12 are the main determinants of circulating homocysteine levels. Due to this reason, these two biomarkers were assessed in the serum samples available of the participants of the Rotterdam Periconception Cohort Study. We would not expect any significant difference in the association between methionine or serine with homocysteine, as they are all part of the same metabolic pathway, just at different steps within the pathway.

Did homocysteine affect fetal development by modulating one-carbon metabolism? Or by other means?

Homocysteine is a known sensitive marker for the functionality of the one-carbon metabolism, which informs us on whether there are sufficient or insufficient substrates or cofactors to adequately drive the one-carbon metabolism and hence supply the pregnancy with substrates for genetic and epigenetic processes. In this case, homocysteine itself is not necessarily the direct cause of adverse fetal development, but a marker for derangements in the pathway. According to the literature, the one-carbon metabolism is fundamental for fetal development, as it provides important substrates for essential cellular processes such as DNA repair and methylation of DNA and histones. 30 years of research shows strong evidence that derangements of the one-carbon metabolism are associated with adverse pregnancy outcomes such as fetal growth alterations and pregnancy complications (examples can be seen in human research from prof. Régine Steegers-Theunissen and animal research from prof. Kevin Sinclair). We hypothesize that one-carbon metabolism derangements, in general reflected by elevated homocysteine levels, may play a major role in prenatal growth.

On the other hand, independent from the one-carbon metabolism, we cannot exclude that elevated homocysteine itself may directly influence and act on biological processes such as apoptosis and hypomethylation which can negatively affect prenatal development as well.

In conclusion, we cannot state if the influence of homocysteine on prenatal development is a direct biological effect or is limited as a marker of one-carbon metabolism derangements. However, in each statistical model used in this study, we adjusted for relevant confounders to take into account the influence of these on measurements of prenatal growth, to better determine and isolate the effect of homocysteine only on prenatal growth.

Please note that track changes in the manuscript include suggestions from both reviewers.

Reviewer 2 Report

In their research article, the Authors aimed to investigate associations between reference range maternal homocysteine and embryonic and fetal growth. From a total of 1060 singleton pregnancies (555 natural and 505 in vitro fertilization/intracytoplasmic sperm injection (IVF/ICSI) pregnancies), they found that homocysteine was negatively associated with first trimester embryonic growth in the included population and, also after stratification for conception mode, this association remained.

The paper has an interesting research question and a strong methodology, readability is good, references appropriate. In the opinion of this reviwer, it deserves to be shared among specialist in human reproduction and feto-maternal medicine. As minor change, I’d like suggest to remove from tables columns of missing values in order to make easy the readability of tables. Additionally, in the discussion section, the clinical implications could be endorsed.

Author Response

As minor change, I’d like suggest to remove from tables columns of missing values in order to make easy the readability of tables. Additionally, in the discussion section, the clinical implications could be endorsed.

Thank you for the suggestions. We adjusted the tables accordingly and extended the paragraph on the clinical implications. The text we added is the following (lines 502-519): ‘Here we confirm again that homocysteine is a relevant biomarker for periconceptional health conditions in clinical practice. Caregivers should be aware that homocysteine is a sensitive marker for the maternal health status, particularly nutrition (B vitamin deficiencies) and lifestyle [2]. We therefore recommend routine analysis of homocysteine levels in preconceptional and pregnant women and their partners, to treat such imbalances in a timely manner, as it also has been reported that high homocysteine levels in men affect sperm quality and DNA methylation [55, 56]. If homocysteine is found elevated during pregnancy, extra fetal ultrasound examinations should be considered to aid the decision making regarding obstetric surveillance (e.g. if growth restriction or changes in growth patterns are detected). Based on our findings, we recommend that special attention should be given to women who require ART treatment and undergo a frozen cycle. Treatment could be posticipated in cases of elevated maternal (and paternal) homocysteine levels. In combination with this, preconceptional and pregnant women should be encouraged to follow a healthy diet and lifestyle, to reduce the risks of elevated homocysteine levels during the periconception period and beyond. This can be achieved via tailored and ‘blended’ lifestyle care(e.g. smarterpregnancy.co.uk)  [57-62].

Please note that the track changes in the manuscript are from suggestions of both reviewers.
